# CRISPR/Cas as a Genome-Editing Technique in Fruit Tree Breeding

**DOI:** 10.3390/ijms242316656

**Published:** 2023-11-23

**Authors:** Marina Martín-Valmaseda, Sama Rahimi Devin, Germán Ortuño-Hernández, Cristian Pérez-Caselles, Sayyed Mohammad Ehsan Mahdavi, Geza Bujdoso, Juan Alfonso Salazar, Pedro Martínez-Gómez, Nuria Alburquerque

**Affiliations:** 1Fruit Biotechnology Group, Department of Plant Breeding, CEBAS-CSIC (Centro de Edafología y Biología Aplicada del Segura-Consejo Superior de Investigaciones Científicas), Campus Universitario Espinardo, E-30100 Murcia, Spaincperez@cebas.csic.es (C.P.-C.); nalbur@cebas.csic.es (N.A.); 2Department of Horticultural Science, College of Agriculture, Shiraz University, Shiraz 7144165186, Iran; sama_rahimi@yahoo.com (S.R.D.); smemahdavi@gmail.com (S.M.E.M.); 3Fruit Breeding Group, Department of Plant Breeding, CEBAS-CSIC (Centro de Edafología y Biología Aplicada del Segura-Consejo Superior de Investigaciones Científicas), Campus Universitario Espinardo, E-30100 Murcia, Spain; gortuno@cebas.csic.es (G.O.-H.); jasalazar@cebas.csic.es (J.A.S.); 4Research Centre for Fruit Growing, Hungarian University of Agriculture and Life Sciences, 1223 Budapest, Hungary; bujdoso.geza@uni-mate.hu

**Keywords:** plants, molecular biology, genomic, transgenic transformation, breeding

## Abstract

CRISPR (short for “Clustered Regularly Interspaced Short Palindromic Repeats”) is a technology that research scientists use to selectively modify the DNA of living organisms. CRISPR was adapted for use in the laboratory from the naturally occurring genome-editing systems found in bacteria. In this work, we reviewed the methods used to introduce CRISPR/Cas-mediated genome editing into fruit species, as well as the impacts of the application of this technology to activate and knock out target genes in different fruit tree species, including on tree development, yield, fruit quality, and tolerance to biotic and abiotic stresses. The application of this gene-editing technology could allow the development of new generations of fruit crops with improved traits by targeting different genetic segments or even could facilitate the introduction of traits into elite cultivars without changing other traits. However, currently, the scarcity of efficient regeneration and transformation protocols in some species, the fact that many of those procedures are genotype-dependent, and the convenience of segregating the transgenic parts of the CRISPR system represent the main handicaps limiting the potential of genetic editing techniques for fruit trees. Finally, the latest news on the legislation and regulations about the use of plants modified using CRISPR/Cas systems has been also discussed.

## 1. Introduction

Fruit trees are crops of great economic importance worldwide as they are vital components of our food production systems. Abiotic stresses such as salinity or drought, the monoculture of disease-susceptible cultivars, excessive use of pesticides, and the appearance of new pathogens cause significant economic losses in the production of various fruit species and are important threats to the environment and to sustainable food production [1]. Fruit trees play an integral role in the food and nutrition industries due to their invaluable primary and secondary metabolites [2]. Fruits are rich sources of dietary fiber, pectin, antioxidant components, phytoestrogens, cyanogenic glucosides, and vitamins that have been recognized for their role in promoting optimal human health and bolstering the body’s defence against illnesses [3].

On the other hand, perennial fruit trees suffer different biological and environmental challenges throughout their life. Specifically, fruit trees are infected by a wide range of pathogenic agents, including fungi, bacteria, and viruses, which can lead to significant economic losses if not properly addressed or managed [1]. Furthermore, in the current climate change scenario that we face, it is increasingly common for fruit trees to not experience enough cold during winter due to increasing temperatures, and are also affected by drought in some areas.

Therefore, there is great interest in obtaining improved fruit varieties with high nutritional quality and resistant to different stresses. Additionally, it becomes imperative to grasp the roles of stress-tolerance-related genes and their regulatory mechanisms for the purpose of developing more resilient varieties. Breeding of fruit crops using conventional means has been effectual in terms of both quality and yield characteristics, although this is a slow breeding method with random consequences due to extrinsic and intrinsic factors such as a long juvenile period, self-incompatibility, heterozygosity, long times for selection of the seedlings, and a lack of correlation between seedlings and mature plants [2]. Traditional breeding methods have been enriched by the inclusion of transgenesis, a valuable tool for plant breeding that enables the introduction or modification of specific and important traits in a single step [4], also allowing functional genomic studies.

Despite their advantages, transgenesis has its own limitations, including the random integration of transgenes into the genome and the fact that many fruit trees species are recalcitrant or time-consuming in their transformation. Therefore, it is paramount to enhance transgenic research and dedicate additional efforts to enhance the efficiency of the regeneration and transformation procedures in fruit trees [5]. Equally, in higher plants, achieving the insertion of DNA sequences at a precise genomic location using homologous recombination, known as gene targeting (GT), has remained challenging due to the notably low efficiency of homologous recombination [6]. One approach to enhancing HR-dependent gene targeting involves inducing double-strand breaks (DSBs) in the genomic DNA at the desired target site [7].

Among the new technologies developed in recent years, various site-specific nucleases (SSNs) have emerged, enabling the precise creation of double-strand breaks (DSBs) at specific locations within the genome. These SSNs offer a highly innovative approach to genome engineering, facilitating targeted modifications such as gene silencing, gene correction, and gene addition [5]. SSNs have significant economic, time-saving, and streamlined advantages relative to conventional breeding methods, which may take up to approximately a decade in order to develop a variety [8]. This methodology can be used to study the genes involved in traits such as drought tolerance, disease resistance, and higher quality and yield [9]. SSNs can be used for different purposes to modify the structure and function of host genome in agricultural crops, such as the targeted mutation, modification, insertion, replacing, stacking, and translational modulation of the desired genes [8].

The classification of SSN-based genome-editing systems is according to the following categories: meganucleases, zinc finger nucleases (ZFNs), transcription-like effector nucleases (TALENs), and Clustered Regularly Interspaced Short Palindromic Repeats (CRISPR), which is associated with the RNA-guided Cas double-stranded DNA-binding protein (CRISPR/Cas system) [5,10]. The main differences among them are their relative specificity and efficiency [11]. ZFNs and TALENs are engineered nucleases and their mode of action is based on the protein–DNA interaction. However, the CRISPR/Cas system depends on RNA–DNA coupling [10]. Even though the use of the synthetic nucleases ZFNs and TALES has allowed the targeting of many genomic sites, the application of these techniques for the edition of plant genomes has been limited [12]. CRISPR is, at this moment, the main technology that research scientists use to selectively modify the DNA of living organisms. CRISPR was adapted for use in the laboratory from naturally occurring genome-editing systems found in bacteria.

Since its discovery, the most used genome-editing tool used in plant research and breeding is CRISPR, associated with protein 9 (CRISPR/Cas9), using a designed RNA-guided Cas9 endonuclease [13]. Specifically, CRISPR/Cas9 from *Streptococcus pyogenes* (SpyCas9) has been successfully used for genome editing in many plant species [14]. However, CRISPR/Cas9 has some inconveniences, such as the limitations of target specificity, activity, efficiency, and targeting scope [15]. These limitations have been overcome by engineering the basic Crispr/Cas9 system and the discovery of other Cas enzymes from various species, extending the range of genome-editing tools [16].

The emergence of CRISPR/Cas technology initiated a new perspective on Genetically Modified Organism (GMO) regulations. The generation of GMOs using transgenesis involves the insertion of foreign DNA into the genome of the plant, which is not allowed in many countries around the world. However, the targeted modification of a gene using CRISPR/Cas technology, producing mutations, is in many cases similar to the application of mutagenic agents that are legally acceptable.

In this review, we have analyzed the methods used to introduce CRISPR/Cas systems into fruit species, as well as the impacts of CRISPR/Cas on altering the plant architecture, improving fruit quality and yield and tolerance to biotic and abiotic stresses. Furthermore, this review includes the latest news on the legislation and regulations about the use of plants modified using CRISPR/Cas systems.

## 2. Mechanism of CRISPR/Cas System

In nature, CRISPR/Cas systems provide prokaryotes with an RNA-guided adaptive immunity against bacteriophages and plasmids [17]. These systems are encoded by the CRISPR array and the accompanying CRISPR-associated (Cas) genes. The CRISPR array contains two types of sequences, palindromic repeats and “spacer” sequences that are derived from a viral or plasmid genome. On the other hand, Cas genes codify different proteins involved in the process [18].

The adaptative immune response consists of three stages: adaptation, expression and maturation, and interference [19]. At the adaptation stage, Cas proteins recognize foreign genetic elements (protospacers) and insert them in between the repeats of the CRISPR array, forming new spacers. The expression and maturation stage consists of the transcription of the CRISPR array into pre-crRNA that is further processed, forming smaller mature crRNAs, one for each spacer. Then, the crRNA forms a complex with Cas proteins, and in some cases with the tracrRNA (transactivating crRNA) [20], which leads to the interference stage. The complex makes differentiations by base-pairing foreign nucleic acids that are complementary to crRNA sequences. In addition, a specific motif called a PAM (protospacer adjacent motif) is necessary for the stable binding to the target DNA, and it is crucial for the discrimination between self and non-self sequences [18]. The recognition of the foreign genetic elements by the Cas–crRNA complex triggers the activation of the nuclease activity of Cas proteins, which degrade the foreign genome, avoiding infection [19].

There is a wide diversity of CRISPR/Cas systems that differ in the Cas protein sequences, gene compositions, and architecture of the genomic loci. According to Makarova et al. [21], CRISPR/Cas systems can be classified into 2 different classes, 6 types, and 33 subtypes, even though this classification is constantly evolving as new systems are being discovered. Class 1 and 2 differ in the number of Cas proteins involved in crRNA processing and interference, class 2 systems being simpler, as they only require one multidomain crRNA-binding protein.

Among class 2 systems, CRISPR/Cas9 and CRISPR/Cas12 are the most used for genome-engineering technologies because of their properties. Both are capable of cleaving dsDNA with just one Cas protein (Cas9/Cas12) being necessary for the recognition and cleavage of the DNA [22]; however, there are some differences in their mechanisms. In both systems, the Cas endonuclease assembles with the crRNA, which binds the target dsDNA by complementarity. For the CRISPR/Cas9 system, tracrRNA is also needed for the processing of crRNA and the interaction between Cas9 and the crRNA. This system has been engineered to create an RNA chimera (sgRNA) that acts as the crRNA and tracrRNA [23]. On the other hand, the CRISPR/Cas12 system does not need the tracrRNA, as it can process its own crRNA [22]. The PAM sequence is also different for each Cas protein. While for Cas9, the PAM sequence is NGG (being N any nucleobase), and it is located in the 3′ end, for Cas12, it is TTTV (V = A, C, and G), and it is located in the 5′ end [24]. Once they have recognized their corresponding PAM sequence, the nuclease activity of the Cas proteins is activated, leading to the dsDNA cleavage which produces a double-strand break (DSB) [25]. Cas9 generates blunt ends at the 3′ while Cas12 creates staggered ends at the 5′ [24]. In both cases, the DSB can trigger two endogenous DNA repair mechanisms: homology-directed repair (HDR) or non-homologous end joining (NHEJ) (Figure 1), both being interesting for genome-engineering applications [25]. HDR occurs if there is a homologous template, useful for changing or replacing sequences [26]. If there is no homologous template, NHEJ is triggered. This mechanism generates small insertions and deletions (indels) in order to ligate the broken ends as fast as possible, leading to the knockout of the gene [26] (Figure 1).

## 3. Genetic Transformation Technology in Fruit Trees

Transformation of several fruit trees has been carried out for many traits and has been improved for a successful genetic transformation so far. Transgenic technology is increasingly used in fruit species to overcome the disadvantages of conventional traditional breeding methods and for gene function research [27]. However, there are several limitations in the transformation of fruit trees. Most of fruit trees are recalcitrant to regeneration and/or transformation, the processes are genotype-dependent, the process is time-consuming compared to for other species, and accurate selection with antibiotics or herbicides is necessary to avoid chimeric plants [5]. Furthermore, the lack of available and efficient explants for regeneration and transformation procedures (e.g., seedlings, leaves from micropropagated plants, or immature seeds) makes difficult the establishment of effective protocols [28]. Moreover, the public concern and the legislative boundaries on GMO production and commercialization hamper the huge biotechnological potential of fruit genetic transformation techniques.

Fruit quality improvement and biotic and abiotic tolerance/resistance have been achieved in fruit scion cultivars using direct transformation, but the use of genetically modified rootstocks to confer new characteristics to the non-transformed scion via transgrafting shows a potential improvement of fruit tree species, in particular those recalcitrant to transformation, and could mitigate public concerns about transgene dispersions or transgenic fruit consumption [28,29]. Additionally, fruit genome editing is a new breeding technology that offers the possibility of producing improved commercial fruit cultivars and could help to address some of the regulatory constraints about the cultivation of first-generation transgenic crops [30].

*Agrobacterium tumefaciens* (*Rhizobium radiobacter*)-mediated transformation has been the prevalent method used for the genetic engineering of fruit tree crops for more than three decades [31]. Successful transformation has been reported for functional genomics studies and the genetic improvement of several fruit crop genotypes by using Agrobacterium infection [28,32].

Although grapevine (*Vitis vinifera*) is considered a recalcitrant specie for transformation [31], over the past few years, numerous works have reported the successful transformation of various grape rootstocks and cultivars using Agrobacterium-mediated and or biolistic bombardment techniques [33]. These transformations have involved a range of target genes, such as genes involved in resistance and tolerance against diseases, pests, and abiotic stresses, as well as enhancing fruit quality [33,34].

In the same way as with grapevine, genetic transformation using Agrobacterium tumefaciens is the most used method to obtain transgenic apple (*Malus domestica*) plants [35]. The genetic modification of apples has been feasible since 1989 [36], and in the following years, most studies were focused on increasing the transformation efficiency. Agrobacterium-mediated transformation has become a conventional tool for functional genome studies on apples using overexpression or RNAi-based gene silencing [35]. However, only a select number of research teams have managed to effectively use transformation methods for breeding purposes. The release of a non-browning transgenic apple, known as Artic^®^ apple, is remarkable, which was developed using sense post-transcriptional silencing of a chimeric polyphenoloxidase gene [37]. The Arctic^®^ apple concept is the result of one breeding program by the biotech company Okanagan Specialty Fruits, and currently there are three commercial varieties of Arctic^®^ apple [30]. Notable advancements, particularly in recent years, have broadened the range of tools available to breeders and researchers involved in breeding apple efforts, like the establishment of protocols using rapid crop cycles in breeding, methods for obtaining marker-free genetically modified plants, or the production of cisgenic apple plants (plants that contain genes present in other species or in cross-compatible relatives, but not foreign genes). Regarding the application of targeted gene silencing, in addition to traditional RNAi-based silencing via stable transformation using hairpin gene constructs, powerful technologies have emerged such as optimized protocols for virus-induced gene silencing (VIGS) and artificial microRNAs (amiRNAs). Furthermore, the establishment of methods for successful targeted genomic editing in apple trees has also been achieved [35].

*Agrobacterium*-mediated transformation of citrus was initially reported by Moore et al. [38] using internodal stem segments as explants, followed by the regeneration of shoots. Extensive research has resulted in the development of improved Agrobacterium protocols for the genetic modification of citrus plants [39,40]. Due to the difficulties of conventional citrus breeding (a complex reproductive biology, juvenility, a high heterozygosity level), genetic transformation has been considered as a possible alternative strategy for citrus improvement [41]. Modified plants from different citrus species have been generated with resistance to diseases such as huanglongbing and citrus canker caused by bacteria [42,43,44] and tristeza disease caused by Citrus tristeza virus [45], as have plants tolerant to different environmental stresses [44].

Although in most woody fruit species and especially in Prunus species, transformation and regeneration are frequently limited to a few genotypes [46], among Prunus, the European plum (*Prunus domestica* L.) is the species most frequently transformed [47]. However, Japanese plum transformation has been reported with low efficiency [48]. In the first works, several marker genes were introduced into the plum genome [49,50]. Also, protocols for alternative selection marker gene introduction and the elimination of marker genes to avoid environmental risks have been reported [51,52]. Regarding the introduced traits in the European plum, resistance against pests, diseases, and/or abiotic stresses, a shorter juvenile period, dwarfing, and continuous flowering have been the main objectives [47].

Sharka disease caused by the plum pox virus (PPV) is the most important disease of stone fruit, and the establishment of new cultivars resistant to sharka is one of the most focused topics in European plum breeding programs [53]. Among the different transgenic strategies used to achieve PPV resistance, successful results have been obtained via applications of RNA silencing techniques [54]. The first PPV-resistant transgenic Prunus was the plum C5 or “Honeysweet”, which was obtained via the Agrobacterium-mediated transformation of plum hypocotyl slices using a binary plasmid carrying the PPV-CP full-length gene [55]. The resistance of “Honeysweet” was due to the post-transcriptional gene silencing (PTGS) of the coat protein (CP) virus gene [56]. “HoneySweet” is freely available for fruit production in the United States and for use as a source of PPV resistance for developing new PPV-resistant plum cultivars worldwide, pending regulatory approval [57]. Different ihpRNA PPV-CP constructs have been designed to obtain new PPV-CP silenced transgenic plum lines [58], and other viral sequences were chosen to induce sharka resistance in the European plum [59], demonstrated to be effective against a wide range of PPV strains [60,61,62]. Recently, the use of PPV-resistant transgenic plum rootstocks has been proposed as a strategy for conferring virus resistance to cultivars or species recalcitrant to transformation, which could mitigate public concerns about transgene dispersion and eating transgenic food [29].

Engineered plum lines were produced via the RNA-interference-mediated silencing of the A. tumefaciens oncogenes ipt and iaaM to study the possibility of generating plum transgenic rootstocks resistant to crown gall disease. Several lines were infected with Agrobacterium strains in the greenhouse, showing a significant reduction in the development of the disease [63].

The use of transgenic Prunus rootstocks resistant to salinity and/or drought could improve productivity in arid and semi-arid regions affected by environmental stresses. Transgenic European plum lines tolerant to salt stress were obtained by overexpressing cytosolic superoxide dismutase (SOD) from spinach and/or cytosolic ascorbate peroxidase (APX) from peas [64,65]. Modulation of the enzymatic antioxidants and enhancement of non-enzymatic antioxidants like glutathione and ascorbate are responsible for the stress tolerance [65]. Additionally, one transgenic line with high APX activity showed tolerance to severe water stress [66].

The European plum has also been transformed with the FLOWERING LOCUS T1 (FT1) gene from Populus trichocarpa, and transgenic plants that expressed high levels of FT1 flowered and produced fruits in the greenhouse within 1 to 10 months [67]. FT plums showed the ability to continuously produce flowers and fruit regardless of the day’s length or chilling time and survived winter temperatures. For these reasons, FT plums are used in crosses at the USDA ARS facility (Kearneysville, WV, USA) in what has been called “FasTrack” breeding [68]. The “FasTrack” system has allowed minimizing the generation cycle of plum plants from 3–7 years to one year round; it can be used under greenhouse conditions and the system allows the fast incorporation of important traits into plums.

Apricot (*Prunus armeniaca* L.) is a very recalcitrant species with important limitations in regeneration and transformation from explants of juvenile or mature origin. There are several works reporting the production of transgenic apricot plants expressing the marker genes gfp or uidA and nptII [69,70,71]. Also, the generation of marker-free transgenic apricot plants was achieved by using the regeneration-promoting gene ipt and site-specific recombination [72] and the chemical-inducible Cre–LoxP system [73].

Although the main goal of transgenic research has been the generation of plants resistant to diseases [74], until now, there have been very few studies indicating the production of transgenic apricot lines with modified target genes for breeding objectives. To this end, Laimer da Câmara Machado et al. [75] produced some transgenic apricot lines with the CP of PPV that showed resistance to viral infection. More recently, Alburquerque et al. [63] reported the generation of engineered apricot plants with the Agrobacterium tumefaciens oncogenes ipt and iaaM silenced, although all transgenic lines were still susceptible to crown gall disease.

Regarding the peach (*Prunus persica* L.), after the first publication reporting the regeneration of transformed plants [76], the lack of efficient genetic transformation protocols has prevented the application of many biotechnological tools in peach breeding programs like RNA interference, cisgenesis/intragenesis, or genome editing [77].

In other fruit crops like papaya (*Carica papaya* L.), important challenges have been achieved. In 1992, papaya ringspot virus (PRSV) was detected in Hawaii. Since the disease caused by PRSV was not completely controlled using conventional methods, local researchers generated transgenic papaya lines that contained the coat protein gene of PRSV, utilizing microprojectile-mediated transformation of immature zygotic embryos of the “Sunset” cultivar. Thus, the transgenic papaya “SunUp” cultivar, which is completely resistant to PRSV, was established [78].

To the best of our knowledge, most gene-editing studies in fruit trees have been performed via Agrobacterium-mediated transformation to stably knock out the genes associated with major agronomic traits, primary and secondary metabolite production, disease resistance, and improved breeding purposes using the popular CRISPR/Cas9 system to achieve editing [79,80]. Table 1 summarizes the results of different gene-editing studies performed with fruit trees indicating the CRISPR/Cas delivery technique.

The polyethylene glycol (PEG)-mediated delivery method has also been employed in fruit tree genome-editing systems (Table 1) since it is especially useful for these species where the production of transgenic plants is very slow [128]. This CRISPR/Cas delivery method has been proposed as a good strategy to produce transgene-free edited plants by delivering ribonucleoprotein [127]. Nevertheless, this method has not been widely used in the genome editing of fruit trees because of low efficiency and the limitations in protoplast preparation, transformation, and regeneration techniques [129]. Malnoy et al. [121] transformed grapevine protoplasts targeted in MLO7 to increase resistance to powdery mildew. PEG-mediated transforming protocols in grapevine have been improved for better editing and protoplast regeneration [124,125]. Also, Malnoy et al. [121] used the same methodology targeting DIPM-1, DIPM-2, and DIPM-4 in apple to increase resistance to fire blight disease. PEG-mediated transformation has carried out in other fruit trees like orange (*Citrus sinensis* (L.) Osbeck) [126,127], banana (*Musa* spp.) [122], and chestnut (*Castanea sativa*) [123].

Another transformation technique to deliver DNA directly into plant cells is particle bombardment (gene-gun-mediated transformation). Microprojectile bombardment is a fast method for delivering the desired genetic materials (DNA, RNA, or ribonucleoproteins) into cells. Unlike Agrobacterium-mediated transformation methods, microprojectile bombardment is not dependent on host specificity or species limitations. This method is the most efficient to achieve organelle transformation, such as for chloroplasts and mitochondria. However, some drawbacks of this direct method are the requirement for expensive equipment and consumables or the frequent complex DNA integration patterns [130]. Moreover, particle bombardment is less efficient than Agrobacterium-mediated methods and it is limited due to worse explant regeneration after treatment and the destruction of genomic sequences as a consequence of the bombardment [32,129]. Although this technique has been established for several fruit tree transformation protocols such as papaya [78], grapevine [34], and citrus [43], to the best of our knowledge, this method is yet to be reported for genome editing in fruit trees.

## 4. CRISPR/Cas-Mediated Gene Knock-In and Knock-Out in Fruit Trees

CRISPR/Cas has been applied to activate and knock out target genes in different fruit tree species, including related to tree development, yield, fruit quality, and tolerance to biotic and abiotic stresses, trying to answer to different challenges (Figure 2).

### 4.1. Tree Growth and Development

The development and growth of the plant is a crucial factor that will determine the size, density, and ultimately the productivity of a plant. To enhance the planting density and subsequently increase productivity while improving nutrient and water use efficiency, the cultivation of dwarf fruit trees has become a prominent strategy [131,132]. Nevertheless, adjusting the plant height in these dwarf crops poses challenges that have led to investigations centered on phytohormones and genetic manipulation.

Phytohormones play pivotal roles in plant growth and architecture [133]. Among these, gibberellins (GAs) are recognized for their ability to stimulate plant elongation [2]. The disruption of genes involved in GA biosynthesis can result in dwarfed plant structures [134]. The *MaGA20ox2* gene is involved in gibberellic acid biosynthesis and plant height in the Gros Michel banana cultivar, and CRISPR/Cas9 technology has been successfully used to modify the *MaGA20ox2* gene and generate semi-dwarf mutants [93]. Given the high conservation of phytohormone pathways across various plant species, researchers have identified several key regulatory genes within these pathways.

In the cytokinin context, Feng et al. [135] observed changes in gene expression associated with the cytokinin metabolic pathway and trans-zeatin concentration in apple rootstocks, distinguishing between vigorous and dwarf variants. They identified decreased expression of the *IPT5b* gene, characterized by high methylation levels in the promoter region, leading to impaired trans-zeatin synthesis and potentially causing dwarfism.

In some instances, abnormal levels of abscisic acid (ABA) can contribute to dwarfism in fruit trees by influencing plant growth and development. Pang et al. [120] supported this concept by generating six homozygous mutant pear lines (*Pyrus betulifolia*) with dwarf phenotypes using *Agrobacterium*-mediated transformation. These mutants exhibited elevated endogenous ABA levels and increased expression of ABA pathway genes, directly linking ABA to dwarfism. Furthermore, as mentioned in Sattar et al. [136], the potential for genetic improvement has been demonstrated in dwarf pears (*Pyrus bretschneideri*) using CRISPR/Cas9 technology, modifying the *PbPAT14* gene function.

Jia et al. [137] conducted research on the *MdNAC1* gene in apples, known to be associated with plant dwarfism. Transgenic plants with overexpressed *MdNAC1* displayed dwarf characteristics, including shorter stems and roots, a reduced leaf area, and decreased levels of brassinosteroids (BR) and ABA. This implies that MdNAC1 regulates dwarfism by affecting BR and ABA production. Subsequently, another follow-up study [138] revealed increased abundance of the MdKNOX15 gene, part of the apple KNOX transcription factor family, in stems of varieties displaying early flowering dwarfism.

Estrigolactone (SL) is a recently identified plant hormone that plays a pivotal role in branching inhibition in plants. Two key genes involved in SL biosynthesis, *CCD7* and *CCD8*, have been investigated in grapevine. CRISPR/Cas9 technology was employed, specifically for editing the *VvCCD7* and *VvCCD8* genes [139]. As a result of these genetic edits, it was observed that the ccd8 mutant exhibited a higher number of branches compared to wild-type plants, highlighting the significance of the *VvCCD8* gene in grapevine branching regulation [129].

Additionally, mutations in the *CNGC2* gene in *Arabidopsis thaliana* have shown pleiotropic effects, including the accumulation of salicylate compounds and the development of dwarf plants [140]. *CNGC2* is also involved in processes such as fertilization and leaf development [141,142], with mutations leading to anomalies in anther length and deficient fertilization. The effectiveness of *MdCNGC2* as a target gene for genetic improvement in apple cultivars allows the application of CRISPR/Cas9 technology to induce mutations, consistently resulting in the accumulation of salicylate compounds in apple calluses, showcasing the versatility of this tool in genetic manipulation [85].

These genes offer promising avenues for genetic editing aimed at creating fruit crops with dwarf phenotypes or modified branch architecture, with significant potential to boost productivity and efficiency in fruit tree agriculture.

### 4.2. Early Flowering

The extended juvenile phase observed in many fruit trees leads to a prolonged non-flowering period, which can span from 3 to 15 years, depending on the specific fruit tree [143]. Elevated levels of terminal flowering protein (TFL) are typically associated with this youthful stage. TFL acts as a negative regulator of flowering by inhibiting the expression of several flowering-stimulating proteins, including FLOWERING LOCUS T (FT), LEAFY (LFY), and APETALA1 (AP1) [144].

To address this challenge, ref. [82] utilized CRISPR/Cas9 technology to edit the apple *TFL1* gene. They employed two different sgRNAs to target the *TFL1* gene, and they also used the same construct to edit the pear *TFL1*. It was observed that early flowering in transgenic pear lines (9%) and transgenic apple lines (93%) targeted the *PcTFL1.1* and *MdTFL1.1* genes, respectively.

Moreover, CRISPR/Cas9 technology was applied to simultaneously target the *AcCEN4* and *AcCEN* genes in kiwi. This transformation changed a perennial climbing plant, which produces axillary inflorescences after years of youth, into a compact plant with rapid terminal flowering, all achieved via mutations in the *CEN* genes [114]. Moreover, in the edited *CEN* lines, early flowering was dependent on the genetic dosage. In other words, plants mutated with all four alleles of *CEN* and *CEN4* exhibited early flowering compared to plants with fewer allelic mutations [2]. The advantage of modifying a perennial plant like kiwi to flower year-round is that it allows for a quick reproductive cycle, producing fruits throughout the year instead of seasonal crops [145].

In trifoliate orange, the overexpression of *Citrus unshiu FLOWERING LOCUS T* (*CiFT*) led to early flowering just 12 weeks after being transferred to the greenhouse [146]. Additionally, the overexpression of FT in *C. excelsa* resulted in early flowering and an accelerated citrus breeding program [147].

The altered expression of key genes that control flower initiation in fruit crops using genome editing holds the potential to reduce the duration of the juvenile phase and expedite genetic improvement [148,149].

### 4.3. Fruit Growth and Development

CRISPR/Cas holds significant potential for fine-tuning valuable quantitative traits in crop improvement, such as fruit size [150]. Nevertheless, it is crucial to emphasize that the manipulation of specific genes can lead to a wide range of effects. For instance, in the case of the *SlKLUH* gene, its copy number has been observed to positively correlate with fruit weight. Nonetheless, the deletion or reduction of *SlKLUH* often results in smaller fruits and can lead to other growth defects, such as smaller inflorescences and sterile flowers [151].

Equally, conserved motifs in the *KLUH* promoter, related to the promotion of plant organ growth, have also been identified in crops such as pepper and sweet cherry. This suggests that the same approach can be used to engineer changes in fruit and seed size using the *KLUH* gene in these crops [152]. These findings underscore the genetic editing potential for enhancing specific traits in agricultural crops, while emphasizing the importance of understanding the associated side effects and genetic intricacies in this process.

### 4.4. Shelf-Life and Fruit Ripening

One crucial aspect of post-harvest fruit quality revolves around its shelf-life, in which ethylene plays a pivotal role in both ripening and fruit softening. The fruit shelf-life can be extended by controlling ethylene biosynthesis and signal transduction, as highlighted in recent studies conducted on apricots and plums using the application of ethylene-related chemicals, either by inhibiting or increasing ethylene production [153].

Fruits with a prolonged shelf-life can be achieved by modifying the methylation patterns or by silencing key genes involved in ethylene biosynthesis, ripening processes, or their signaling pathways in fruit crops.

To illustrate, the CRISPR/Cas9 gene-editing technique was employed to eliminate *MaACO1* (aminocyclopropane-1-carboxylate oxidase 1), the gene responsible for encoding the enzyme that converts ACC into ethylene. This genetic modification resulted in an extended fruit shelf-life period of up to 40 days when compared to wild-type bananas [154]. Moreover, the edited *MA-ACO1* lines yielded fruits with an increased vitamin C content [155].

### 4.5. Fruit Color, Flavor, and Bioactive Compounds

Genomic editing of key genes offers the potential to design fruit crops with elevated levels of functional metabolites and pigments, which can have a significant impact on enhancing the quality and nutritional characteristics of fruits. Red fruits, rich in various bioactive components and nutrients such as antioxidants, minerals, vitamins, and dietary fiber, have been the subject of research [156]. In this context, CRISPR/Cas9 technology has demonstrated its ability to modulate specific traits in fruits. For instance, editing of *MdMKK9* using this technology resulted in a positive increase in anthocyanin expression levels, leading to the characteristic red color in apples [87].

Furthermore, plants can produce various secondary metabolites that play a significant role in growth control, component regeneration, and nutrient enhancement. Carotenoids, for instance, participate in processes like photosynthesis and have antioxidant functions, contributing to attractive colors in fruits and other plant organs [157]. Phytoene desaturase (PDS) has been identified as the rate-limiting enzyme in the carotenoid synthesis pathway [129]. Eliminating PDS using the CRISPR/Cas9 system resulted in mutants with an albino phenotype, underscoring the fundamental importance of this gene in carotenoid synthesis.

Enrichment of bananas with β-carotene was achieved by modifying the lycopene epsilon cyclase (*LCYε*) gene using CRISPR/Cas9 [136]. Sequence analysis revealed insertion–deletion mutations in the *LCYε* gene, significantly increasing β-carotene accumulation without adversely affecting the agromorphological characteristics. In the case of pomegranates, a fruit rich in phenolic metabolites, two UDP-glycosyltransferase genes, *PgUGT84A23* and *PgUGT84A24*, were deleted using CRISPR/Cas9, resulting in a reduction in phenolic anthocyanin content [158].

Additionally, the regulation of tartaric acid (TA) biosynthesis in grapes involves the L-idonate dehydrogenase (*IdnDH*) gene, which was genetically eliminated using CRISPR/Cas9 [105]. In apples, the *MdAAT1* gene has been identified as responsible for ester synthesis [159]. Furthermore, a similar regulatory role of *NAC* transcription factors in flavor ester formation in apples highlights the importance of *NAC1* homologs in different fruit species [160].

### 4.6. Improving Stress Tolerance in Fruit Trees

Abiotic and biotic stresses, such as drought, extreme temperatures, pests, and diseases, pose substantial threats to global food production. These challenges can result in reduced crop yields, lower crop quality, and economic losses for farmers. To mitigate these issues, the advent of gene-editing systems has opened up promising avenues for agriculture. Gene-editing techniques like CRISPR/Cas9 enable precise modifications to an organism’s DNA, allowing researchers and breeders to target specific genes associated with stress resistance or desired traits [161].

Nowadays, CRISPR/Cas technology has been harnessed to confer tolerance to various environmental stresses, as demonstrated in recent studies [162]. Within this context, it has been observed that the *Dehydration-Responsive Element-Binding* (*DREB*) transcription factor (TF) plays a pivotal role in the regulation of several stress-inducible genes. It has been substantiated that *DREB2*-type proteins, a subtype of *DREB* proteins, play a significant role in enhancing drought, salinity, and heat tolerance in a variety of plants, including fruit-bearing trees [163]. For instance, the overexpression of *MsDREB6.2* has notably ameliorated drought resistance in transgenic apple lines [164], while the overexpression of AtDREB1b, a cold-inducible TF, has bolstered cold resistance in transgenic grapevines [165].

CRISPR/Cas9 has evolved into an effective tool for introducing robust transcriptional regulatory elements into the promoter region of genes, governing the expression of stress-responsive genes. Consequently, this augmentation enhances their expression, thereby fortifying plant stress resilience.

The modulation of stomatal activity in response to abscisic acid (ABA) through the *AtMYB60* gene in *A. thaliana* has been shown to exert an influence on drought resistance [166]. In an endeavor to enhance drought resistance in grapevine, CRISPR/Cas9 has been employed to mutate the ortholog of *AtMYB60*, known as the *VvMYB60* gene [83].

Moreover, in related research, the overexpression of the apple spermidine synthase gene has bestowed multiple types of resistance against environmental stressors in pear trees [167].

The involvement of the *Calcium-Dependent Protein Kinase* (*CDPK*) in response to environmental stressors has been extensively documented by Wang and their team [168]. Overexpression of the apple *CDPK* gene, referred to as *MdCIPK6L*, has substantially heightened resistance to saline, osmotic, drought, and cold stresses without compromising root growth [169]. Furthermore, a reduction in *MdNPR1* expression has been discerned in apple plants in response to drought conditions [170].

The overexpression of drought-responsive genes, achieved via genetic editing via the CRISPR/Cas9 system, holds promising prospects for the cultivation of high-yield, high-quality fruit trees in arid regions, thereby contributing to the global food supply for the populace [171].

Editing genes that negatively regulate plant immunity is a strategy for obtaining disease-resistant crops. A notable example of this is the elimination of the *NPR3* gene, a suppressor of defense responses, which has enhanced the resistance of cacao leaves to the *Phytophthora tropicalis* pathogen [96]. Furthermore, the overexpression of *AtNPR1* or its counterparts in other plants such as *A. thaliana* [172], grapevine [173], and apple [174] has demonstrated increased disease tolerance.

In the specific case of the disease caused by the fungus *Botrytis cinerea*, the *VvWRKY52* gene has been identified as being linked to the biological stress response. Utilizing CRISPR/Cas9 technology, this gene was successfully removed in grapevines, resulting in a significant improvement in tolerance to *Botrytis cinerea* [129].

On another note, citrus canker disease caused by *Xanthomonas* spp. poses a serious threat to citrus crops. The *CsLOB1* gene has been pinpointed as a key player in citrus canker resistance in Duncan grapes. Using CRISPR/Cas9, this gene was modified in this species, showing varying levels of resistance to citrus canker [99].

Genes from the *MLO* (*MILDEW RESISTANCE LOCUS*) family have been associated with susceptibility to powdery mildew in several plant species [175]. Inactivating these genes has led to increased tolerance to powdery mildew in various species. For instance, mutation of the *VvMLO3* [111] and *VvMLO7* alleles [83] using CRISPR/Cas9 has conferred greater powdery mildew resistance in grapevines. Furthermore, Dalla Costa et al. [83] designed vectors carrying the CRISPR/Cas9 system to knock out the genes *MdDIPM1* and *MdDIPM4*, which have a role in the susceptibility to fire blight in apple and evaluated some strategies to eliminate exogenous DNA via site-specific removal mechanisms. The elimination of the *EIF4E* gene has improved papaya’s immunity against the papaya ring spot mosaic virus [176], and the removal of the *MdCNGC2* gene in apples has increased resistance to *B. dothidea* [85]. This underscores the effectiveness of genome editing, utilizing the CRISPR/Cas9 system, in enhancing the resistance of various plants to diseases.

## 5. Off-Target Issues

One of the main advantages of CRISPR-Cas systems is their specificity; however, some off-target editing events can occur, leading to undesired modifications. Cas nuclease activity can be triggered even if there is an imperfect complementarity between sgRNA and the off-target genomic site [177], mainly when the mismatches are located far from the PAM sequence [178].

The presence of off-target mutation can be analyzed by sequencing potential off-target sites with a variable number of mismatches. Although it is important to explore off-target effects, there are few studies in fruit trees that address this aspect. In those where it has been carried out, it does not seem to be a critical factor. Thus, in sweet orange, no off-target mutations at potential sites were detected [24,104]. Also, in grapevine, only a fortuitous off-target mutation was observed in one of two edited genes [110].

In order to decrease off-target modifications if they are a problem, some strategies can be followed. First, it is important to design highly specific gRNAs. For this, it is possible to use truncated sgRNAs (tru-gRNAs) that are formed of sequences shorter than 20 nucleotides, which reduces the likelihood of complementarity with mismatches while maintaining on-target efficiency [179]. In *A. thaliana*, tru-gRNAs have been used, resulting in no off-target modifications [180].

Cas nucleases also play an important role in specificity. Different natural variants of Cas nucleases that come from different microorganisms can identify diverse PAM sequences. For example, SaCas9 (from *Staphylococcus aureus*) has been used to obtain higher on-target efficiency and less off-targets compared to SpCas9 (from *Streptococcus pyogenes*) in *A. thaliana*. The higher specificity of SaCas9 is due to the requirement for a longer PAM sequence. However, this also results in a lower number of potential target sites [181]. It is also possible to improve Cas nuclease specificity using protein engineering. Slaymaker et al. [178] modified amino acids that are crucial for the interaction between the sgRNA–Cas and the target sequence, obtaining a more specific Cas called eSpCas9 (enhanced specificity) with great results in rice [182].

Another approach is to use paired nCas9 (nickase Cas9). nCas9 is mutated in one of its nuclease domains; therefore, it makes a single-stranded break in the DNA. This way, two nCas that are leaded by two paired sgRNAs recognizing the target sequence are necessary to create a double-stranded break, doubling the recognition site [183].

Although different strategies have been described to minimize the problem of off-targets, so far, they have not been used in fruit trees. Therefore, it would be interesting to explore its application to this type of crops.

## 6. Regulatory Limits of Genome Editing

Despite the numerous advantages that genome-editing technologies offer to obtain improved fruit trees compared to conventional breeding techniques [136], the legislative limitations that regulate agricultural production have frequently hampered the important potential of these biotechnological tools [83]. The regulatory regimes applied to GMOs in each country are different, being more permissive in North and South America, Australia, and certain parts of Asia. Other countries such as Japan, New Zealand, Norway, Switzerland, and the European Union (EU) have set up more restrictive regimes, and the number of approvals for GMO cultivation and commercial use has been strictly limited [184]. In particular, the current EU legislation on GMOs and derived products for food is based on Directive 2001/18/EC and Regulations 1829/2003 and 1830/2003, which indicate that the authorization regime of a GMO event requires an environmental and human health risk assessment.

The New Genomic Techniques (NGTs), which include targeted mutagenesis and cisgenesis or intragenesis, provide new opportunities to alter the genetic material in a different way from established genomic techniques, with higher precision and speed in introducing the chosen genetic modifications only from a crossable species. Certain targeted mutated crops are indistinguishable from the original plant cultivar, natural mutations, or from genetic modifications introduced by conventional breeding techniques.

Nowadays, an increasing number of countries have adapted the regulatory status of genome-edited plants, by releasing some of the editing technologies from the conventional GMO regulations [185]. This is the case for important exporters of agricultural products countries such as the United States, Canada, Brazil, Argentina, and Australia [184,186].

On the other hand, in July 2018, the European Court of Justice (ECJ) ruled that plants obtained by NTGs are to be considered GMOs and as such must comply with the regulations contained in Directive 2001/18/EC concerning the approval of GMOs [83,185]. However, the organisms obtained using conventional chemical- or radiation-induced random mutagenesis methods are excluded from the scope of the Directive [187]. After the ECJ ruling, a policy debate within the EU was initiated. In 2019, the Council of the European Union requested the European Commission to conduct a study on the impact of the ECJ ruling considering the technical status of novel genomic techniques, ethics, and the views of the EU countries and stakeholders [187]. In 2021, this study concluded that the EU legislation is not fit to regulate plants obtained with some NGTs and that the current legislation needs to be adapted to scientific and technical progress in this area [184,185]. Based on this study, the European Commission (EC) suggested a revision and promoted citizen, stakeholder, and Member State consultations in 2022.

Recently, as a result of this consultation, the EC published a proposal (5 July 2023) for a “REGULATION OF THE EUROPEAN PARLIAMENT AND OF THE COUNCIL on plants obtained by certain new genomic techniques and their food and feed, and amending Regulation (EU) 2017/625”. This proposal of the regulation only affects plants obtained using targeted mutagenesis and cisgenesis (including intragenesis) that do not carry genetic material from non-crossable species, and establishes two categories of plants obtained by NGTs: plants comparable to naturally occurring or conventional plants (category 1 NTG), and plants with modifications that are more complex (category 2 NTG). The category 1 NGT plants and their progeny obtained using conventional breeding techniques have comparable risks to conventionally bred plants, and therefore it is proposed to completely derogate from the European Union legislation on GMOs and GMO-related requirements in sectoral legislation. The recent regulation must be adopted by the EU Member States in the Council and the European Parliament, following the ordinary legislative procedure, to become law.

Genetically edited plants that can benefit from this possible law must not include any element of the CRISPR systems; therefore, the production of transgene-free genome-edited plants is paramount, particularly in the case of fruit crops, due to their long growth cycle, which makes difficult the elimination of CRISPR elements using successive crossing. Although there are several methods to generate these types of plants, like delivering CRISPR components in the form of mRNA or ribonucleoprotein complexes, the use of mRNA or ribonucleoprotein biolistic delivery, and PEG-mediated transformation followed by protoplast regeneration [188], they have been successfully reported only in herbaceous crops [189,190,191,192].

## 7. Conclusions

Successful genome-editing studies on fruit trees show that CRISPR/Cas can induce changes in target genes. In the future, the application of gene editing could allow the development of new generations of fruit crops with improved traits by targeting different genetic segments or even could facilitate the introduction of traits into elite cultivars without changing other traits. Although the use of genome-editing techniques promises a quick, easy, and inexpensive way to develop novel crop varieties with improved traits compared to before, in the case of fruit crops, the scarcity of efficient regeneration and transformation protocols in some species, the fact that many of these procedures are genotype-dependent, their polyploidy, and the convenience of segregating the transgenic parts of the CRISPR system are some of the problems that limit the potential of genetic editing techniques for fruit trees. More research is needed to establish efficient regeneration and transformation protocols for fruit tree species. Transgrafting techniques using CRISPR transgenic rootstocks with resistance to root diseases, drought, or salt stress for the transmission of desired traits to scions may be an alternative strategy to obtain non-transgenic fruit with improved characteristics. The methods for producing transgene-free genome-edited plants or those that avoid the need to remove CRISPR DNA from edited plants still require lengthy tissue culture, and they are not applicable to many species recalcitrant to genetic transformation and regeneration, including most fruit trees. Therefore, the optimization and adaptation of these methods is a challenge for these crops. Finally, a new perspective from society together with possible changes in the legislative framework pave the way to raise public and regulatory concerns about the use of NTG plants, including fruit crops.

## Figures and Tables

**Figure 1 ijms-24-16656-f001:**
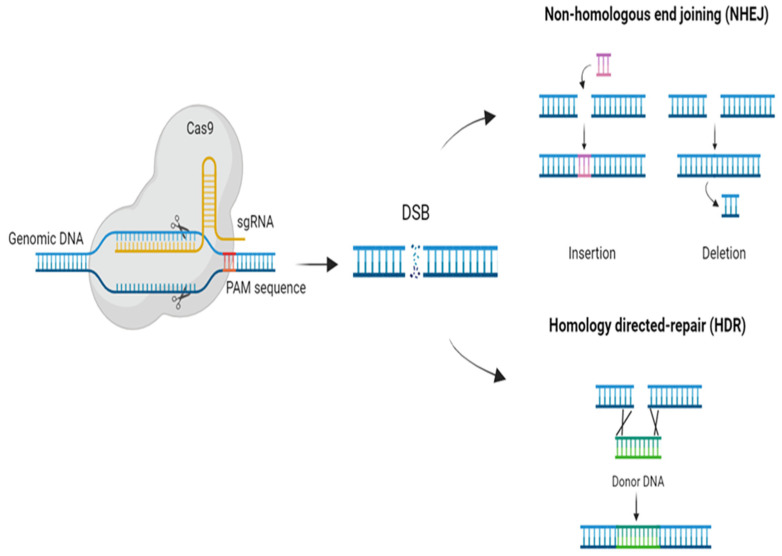
Gene-editing mechanism of CRISP/Cas9. A single RNA chimera (sgRNA) drives the complex CRISPR/Cas9 to the target DNA and the protospacer adjacent motif (PAM) enables the stable binding. Cas9 produces a double-strand break (DSB) that can trigger two endogenous DNA repair mechanisms: homology-directed repair (HDR) or non-homologous end joining (NHEJ).

**Figure 2 ijms-24-16656-f002:**
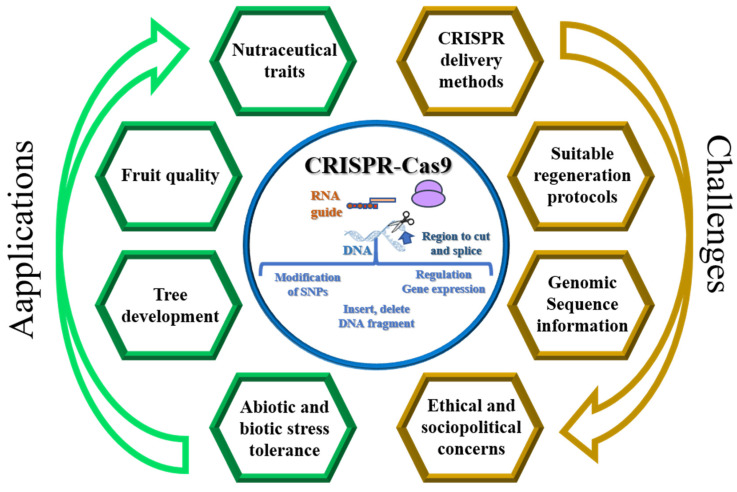
Overview of the applications and challenges of the CRISPR-Cas9 gene-editing technique in fruit trees.

**Table 1 ijms-24-16656-t001:** Collection of traits improved by CRISPR/Cas9 and CRISPR/Cas12 in different fruit trees.

CRISPR-Cas Delivery	Fruit Trees	Target Gene Edited	Improved Traits	References
*A. tumefaciens*-mediated	Apple	*PDS*	Enhanced biosynthesis of carotenoid	[81]
		*PDS* and *TFL1*	Albino phenotype and early flowering	[82]
		*MdDIPM1* and *MdDIPM4*	Fire blight resistance	[83]
		*ALS*	Chlorsulfuron resistance	[84]
		*CNGC2*	*B. dothidea* resistance	[85]
		*MsPDS*	Albino phenotype	[86]
		*MdMKK9*	Increased anthocyanin content	[87]
			Detection of viruses and viroids with CRISPR	[88]
	Banana	*RAS-PDS1* and *2*	Albino phenotype	[89]
		*PDS*	Albino phenotype and dwarfing	[90,91]
		*eBSV*	Control of virus pathogenesis	[92]
		*MaGA20ox2*	Dwarf phenotype	[93]
		*LCYε*	Carotene biosynthesis	[94]
		*MaACO1*	Fruit ripening	[95]
	Cacao	*TcNPR3*	*Phytophthora tropicalis* resistance	[96]
	Citrus	*CsPDS*	Method optimization	[97]
		*Cs7g03360*	Phenotypic changes in Carrizo leaves	[98]
		*CsLOB1*	Citrus canker resistance	[99]
		*CsLOB1* promoter	Citrus canker resistance	[100]
		*PDS* and *CsLOB1*	Albino phenotype and canker resistance	[24]
		*CsWRKY22*	Citrus canker resistance	[101]
		*CsLOB1*	Citrus canker resistance	[102]
		*pC-PDS1* and *Pc-PDS2*	Chlorophyll and carotenoid content	[103]
		*CsLOB1*	Citrus canker resistance	[104]
	Grape	*IdnDH*	High tartaric acid biosynthesis	[105]
		*VvPDS*	Increased carotenoid biosynthesis	[106]
		*VvWRKY52*	*Botrytis cinerea* resistance	[107]
		*VvPDS*	Albino phenotype	[108]
		*VvMLO7*	Powdery mildew resistance	[83]
		*VvPR4b*	Downy mildew resistance	[109]
		*MYBA5/6/7* and *TAS4a/b*	Anthocyanin accumulation	[110]
		*VvMLO3* and *VvMLO4*	Powdery mildew resistance	[111]
		*TMT1* and *TMT2*	Reduced sugar accumulation	[112]
		*TMT1* and *DFR1*	Flavonoid accumulation	[113]
	Kiwifruit	*CEN4* and *CEN*	Terminal flower and fruit	[114]
		*AcPDS*	Albino phenotype (leaves)	[115]
		*AcBFT2*	Reduced dormancy and early bud break	[116]
	Papaya	*CpDreb2*	Gene disruption for water stress	[117]
		*PpalEPIC8*	*Phytophthora palmivora* resistance	[118]
		*Ppal15kDa*	*Phytophthora palmivora* resistance	[119]
	Pear	*TFL1*	Early flowering	[82]
		*PbPAT14*	Dwarf yellowing phenotype	[120]
		*PDS* and *ALS*	Albino phenotype and chlorsulfuron resistance	[84]
PEG-meditated	Apple	*DIPM-1, -2* and *-4*	Fire blight resistance	[121]
	Banana	*PDS*	Method optimization	[122]
	Chestnut	*PDS*	Method optimization	[123]
	Grape	*MLO7*	Enhanced biotic resistance against powdery mildew	[121]
		*DMR6* and *MLO6*	Downy and powdery mildew resistance	[124]
		*GFP*	Method optimization	[125]
	Orange	*CsNPR3*	Induced biotic stress tolerance	[126]
		*PH5*	Method optimization	[127]

## Data Availability

The data are contained within the manuscript.

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
