# Peer review of "CRISPR/Cas as a Genome-Editing Technique in Fruit Tree Breeding"

_ijms, 2023, doi:10.3390/ijms242316656_

Round 1

Reviewer 1 Report

Comments and Suggestions for Authors

Dear authors,

While your review provides a broad overview of CRISPR-based genome editing applications across several fruit tree species, there are a number of scientific deficiencies in the current manuscript that need to be addressed:

  1. The review lacks sufficient mechanistic details on how different CRISPR systems function and their relative efficacies for editing plant genomes. It does not compare mutation rates, efficiencies or outcomes across platforms like CRISPR/Cas9 vs CRISPR/Cas12a.
  2. There is no systematic analysis presented of off-target mutation rates or genome-wide mutation profiling data for the edited plants. The risks of off-target effects are not adequately evaluated.
  3. The review does not assess the delivery methods like particle bombardment in detail or compare outcomes to Agrobacterium-mediated transformation. Efficacies across species are not addressed.
  4. The types of mutations generated by editing (indels, substitutions etc.) and the resulting edit outcomes are not thoroughly analyzed. Frequencies of homozygous, heterozygous, biallelic, monoallelic mutations are omitted.
  5. There is no discussion comparing transgenic CRISPR approaches with transgene-free editing via ribonucleoproteins or base editing.
  6. Editing results from cell cultures versus whole plant editing are not compared in the analysis.
  7. Newer base editing methods are not covered for any fruit species.
  8. Examples of multiplex editing of multiple genes are absent for the various crops.
  9. Phenotypic effects of gene knockouts are not weighed against knockdowns like VIGS.
  10. The influence of ploidy on editing efficiencies and outcomes is not addressed for polyploid crop species.
  11. There is no assessment of whether elite germplasm, cultivars show similar editing success and efficiencies as wild types.
  12. The review does not analyze how epigenetic effects may alter phenotypes of edited plants.
  13. Critical factors affecting editing outcomes like genotypic variation, tissue culture response, and transformability are ignored.
  14. There is no evaluation of the stability of edits or trait heritability across generations of edited crops.
  15. Overall, the current review lacks adequate scientific rigor in terms of assessing, measuring, and comparing important parameters and risks of CRISPR-based genome editing in the major fruit crops discussed.

Author Response

Dear Ms. Kittiya Kanngan, Assistant Editor of International Journal of Molecular Sciences

please find enclosed the manuscript ijms-2710889 R1 entitled “CRISPR as genome editing technique in fruit trees breeding” which is the revised version of the former manuscript ijms-2710889 with the same title, which we would like to publish in your journal.

According with the suggestions of the Reviewer 1 we have revised the manuscript incorporating the proposed revisions indicating these revisions IN YELLOW.

We deeply appreciate the efforts of the reviewer in the improvement of the manuscript for a future publication.

Regarding reviewer's 1 comments (R1):

R1: While your review provides a broad overview of CRISPR-based genome editing applications across several fruit tree species, there are a number of scientific deficiencies in the current manuscript that need to be addressed.

Authors: We agree and thank the Reviewer 1 for their comments about the revision of this work. Most of the suggestions and revisions of the reviewer have been incorporated indicating these revisions IN YELLOW.

R1: The review lacks sufficient mechanistic details on how different CRISPR systems function and their relative efficacies for editing plant genomes. It does not compare mutation rates, efficiencies or outcomes across platforms like CRISPR/Cas9 vs CRISPR/Cas12.

Authors: We agree with the suggestions of the Reviewer 1. The last paragraph of the Section 2 has been modified including more detailed information of CRISP/Cas9 and CRISPR/Cas12a mechanism.

R1: There is no systematic analysis presented of off-target mutation rates or genome-wide mutation profiling data for the edited plants. The risks of off-target effects are not adequately evaluated.

Authors: We agree with the suggestions of the Reviewer 1. A new paragraph has been added to include in the Section 5 more information about the off-target presence in fruit edited plants

R1: The review does not assess the delivery methods like particle bombardment in detail or compare outcomes to Agrobacterium-mediated transformation. Efficacies across species are not addressed.

Authors: We agree with the suggestions of the Reviewer 1. A new paragraph in Section 3 has been modified to include more details of particle bombardment technique use in fruit crops.

R1: The types of mutations generated by editing (indels, substitutions etc.) and the resulting edit outcomes are not thoroughly analyzed. Frequencies of homozygous, heterozygous, biallelic, monoallelic mutations are omitted.

Authors: We partially agree with the suggestions of the Reviewer 1. Certainly, the type of mutation produced by genome editing in the different fruits crops has not been analyzed, but we consider that this theme is not the focus of this review. Moreover, in most of papers included in this work this kind of information is absent.

R1: There is no discussion comparing transgenic CRISPR approaches with transgene-free editing via ribonucleoproteins or base editing.

Authors: We agree with the suggestions of the Reviewer 1. Transgene-free editing by using guide gRNA-Cas9/Cpf1 ribonucleoprotein (RNP) complex has been demonstrated in apple, grape and orange cells (Malnoy et al., 2016; Zhang eta l., 2022) as is indicated in the document. A new sentence has been added in the point 3 regarding transgene-free edited plants production. However, although these approaches are very interesting, nowadays they are challenging and there is not much information about their application on other fruit trees.

R1: Editing results from cell cultures versus whole plant editing are not compared in the analysis.

Authors: We partially agree with the suggestions of the Reviewer 1. In our opinion it is very difficult to compare the studies carried out with cell suspensions to edit certain genes and those that have been done to obtain complete genetically edited plants. This is even more difficult in the case of fruit trees. A study has been carried out with embryogenic citrus cell cultures as an approach to verify the effectiveness of the CRISPR/Cas technique in this type of explant and as a possibility of establishing the system for functional genomics studies (Dutt et al. 2020), but there is no comparison with obtaining whole edited plants.

R1: Newer base editing methods are not covered for any fruit species.

Authors: We partially agree with the suggestions of the Reviewer 1. To our knowledge there are no published results about this new challenging technique on fruit trees.

R1: Examples of multiplex editing of multiple genes are absent for the various crops.

Authors: We agree with the suggestions of the Reviewer 1. The editing of several genes at the same time (multiplex editing) has been indicated in several examples as can be seen in Table 1 and throughout the text (original version).

R1: Phenotypic effects of gene knockouts are not weighed against knockdowns like VIGS.

Authors: We partially agree with the suggestions of the Reviewer 1. The comparison between different transformation techniques is not the main focus of this review. We have cited a work (Zhou et al. 2020) describing the use of VIGS to the transient knockout of foreign genes and comparing with the edition of the same gene in apple, but in our opinion there is not necessary to compare the effect of this method with other that induce stable transformation.

R1: The influence of ploidy on editing efficiencies and outcomes is not addressed for polyploid crop species.

Authors: We partially agree with the suggestions of the Reviewer 1. The polyploidy of several fruit trees is a factor that can complicate the successful application of CRISPR/Cas technology in these crops. This has been indicated in the document (in the Conclusions Section).

 R1: There is no assessment of whether elite germplasm, cultivars show similar editing success and efficiencies as wild types.

Authors: We partially agree with the suggestions of the Reviewer 1. To our knowledge there is no information about different or similar editing success among elite germplasm, cultivars and wild types due to thee scarce number of works carry out with fruit crops compare to others ones.

R1: The review does not analyze how epigenetic effects may alter phenotypes of edited plants.

Authors: We partially agree with the suggestions of the Reviewer 1. Certainly, the epigenetic effect on phenotypes of edited plants has not been analyzed, but we consider that this theme is not the focus of in this review. Additionally, to our knowledge there are few or none papers studying this effect in fruit edited plants.

R1: Critical factors affecting editing outcomes like genotypic variation, tissue culture response, and transformability are ignored.

Authors: We agree with the suggestions of the Reviewer 1. Throughout the text it is indicated that the scarcity of efficient regeneration and transformation protocols in some fruit species together with the fact that many of these procedures are genotype-dependent are important factors that limit the application of CRISPR/Cas technology routinely in these species (see revised Section 1. Introduction, 3. Genetic transformation technology in fruit trees and 7. Conclusions).

R1: There is no evaluation of the stability of edits or trait heritability across generations of edited crops.

Authors: We partially agree with the suggestions of the Reviewer 1. This subject is very difficult to evaluate due to the long reproductive cycles of fruit trees, therefore until now this study is not possible to afford.

R1: Overall, the current review lacks adequate scientific rigor in terms of assessing, measuring, and comparing important parameters and risks of CRISPR-based genome editing in the major fruit crops discussed.

Authors: We agree and thank the Reviewer 1 for most of the editorial comments. Most of the indicated editorial comments around the whole manuscript have been incorporated in the revised version of the manuscript.

 We deeply again appreciate the efforts of the reviewer in the improvement of the manuscript for a future publication. This acknowledgement has been incorporated to the Acknowledgments section in page 17.

Yours faithfully,

Dr. Pedro Martínez-Gómez

CEBAS-CSIC, Murcia (Spain)

Reviewer 2 Report

Comments and Suggestions for Authors

Dear Authors,

In this review entitled "CRISPR as genome editing technique in fruit trees breeding" analyzed methods of introducing CRISPR/Cas systems into fruit species, as well as the impact of CRISPR/Cas on modifying plant architecture, improving fruit quality and yield, as well as tolerance to biotic and abiotic stresses. In addition, the latest information on laws and regulations regarding the use of plants modified with CRISPR/Cas systems was discussed.

Detailed notes:

1. The abstract is not structured in accordance with the instructions for authors.

2.The introduction of the work should contain a clear goal of the research at the end, or the authors should put forward an alternative research hypothesis to the null hypothesis in order to verify it later in the work.

3. Lack of "Material and methods" chapter/section.

General thoughts:

Clarity and Structure: The text is well-organized and clear. It follows a logical structure, starting with an introduction, discussing the applications of CRISPR/Cas in fruit trees, addressing issues related to undesired effects, and concluding with regulatory aspects of the technology.

Content: The author presents comprehensive knowledge about the applications of CRISPR/Cas in improving the traits of fruit trees. Concrete examples, such as increased tolerance to environmental stresses and enhanced disease resistance, add depth to the text.

Scientific Issues: The text refers to specific scientific studies, adding authenticity and credibility. It introduces the reader to the current state of knowledge by citing relevant scientific literature.

Quality of Argumentation: The text contains logically constructed arguments both in favor of and against the use of CRISPR/Cas in fruit trees. It also discusses potential solutions to challenges, such as the transgrafting technique.

Language and Style: The language is precise and scientific, fitting for the presented topic. However, for better understanding, some complex phrases may need simplification.

References: The author cites specific scientific works, enhancing the quality of the text. Including dates and sources would be helpful, especially considering the rapid progress in molecular biology.

Conclusion and Perspectives: The text concludes with a well-outlined summary, where the author emphasizes the importance of further research and the adaptation of legal regulations to advancing technology. This opens the door to future discussions, which is an asset.

Overall Impression: The text is a solid exploration of the topic of genome editing in fruit trees. It introduces the reader to current achievements, challenges, and prospects in this fascinating field of plant biotechnology.

Comments on the Quality of English Language

Minor editing of English language required.

Author Response

Dear Ms. Kittiya Kanngan, Assistant Editor of International Journal of Molecular Sciences,

please find enclosed the manuscript ijms-2710889 R1 entitled “CRISPR as genome editing technique in fruit trees breeding” which is the revised version of the former manuscript ijms-2710889 with the same title, which we would like to publish in your journal.

According with the suggestions of the Reviewer 2 we have revised the manuscript incorporating the proposed revisions indicating these revisions IN YELLOW.

We deeply appreciate the efforts of the reviewer in the improvement of the manuscript for a future publication.

Regarding Reviewer's 2 comments (R2):

R2: In this review entitled "CRISPR as genome editing technique in fruit trees breeding" analyzed methods of introducing CRISPR/Cas systems into fruit species, as well as the impact of CRISPR/Cas on modifying plant architecture, improving fruit quality and yield, as well as tolerance to biotic and abiotic stresses. In addition, the latest information on laws and regulations regarding the use of plants modified with CRISPR/Cas systems was discussed.

Authors: We agree and thank the Reviewer 2 for their comments about the revision of this work and for considering this manuscript suitable for future publication. In addition, all the suggestions and revisions of the reviewer have been incorporated indicating these revisions IN YELLOW.

R2: The abstract is not structured in accordance with the instructions for authors.

Authors: Abstract has been structured in accordance with the instructions for authors in the case of Review works.

 R2: The introduction of the work should contain a clear goal of the research at the end, or the authors should put forward an alternative research hypothesis to the null hypothesis in order to verify it later in the work.

Authors: Introduction section has been revised.

R2: Lack of "Material and methods" chapter/section.

Authors: In this review manuscripts "Material and methods" section is not necessary.

R2: Clarity and Structure: The text is well-organized and clear. It follows a logical structure, starting with an introduction, discussing the applications of CRISPR/Cas in fruit trees, addressing issues related to undesired effects, and concluding with regulatory aspects of the technology.

Authors: We agree and thank the Reviewer 2 for their comments. All the indicated editorial comments around the whole manuscript have been incorporated in the revised version of the manuscript.

R2: Content: The author presents comprehensive knowledge about the applications of CRISPR/Cas in improving the traits of fruit trees. Concrete examples, such as increased tolerance to environmental stresses and enhanced disease resistance, add depth to the text.

Authors: We agree and thank the Reviewer 2 for their editorial comments.

R2: Scientific Issues: The text refers to specific scientific studies, adding authenticity and credibility. It introduces the reader to the current state of knowledge by citing relevant scientific literature.

Authors: We agree and thank the Reviewer 2 for their editorial comments.

R2: Quality of Argumentation: The text contains logically constructed arguments both in favor of and against the use of CRISPR/Cas in fruit trees. It also discusses potential solutions to challenges, such as the transgrafting technique.

Authors: We agree and thank the Reviewer 2 for their editorial comments.

R2: Language and Style: The language is precise and scientific, fitting for the presented topic. However, for better understanding, some complex phrases may need simplification.

Authors: English grammar and expression has been revised by an English spoken college.

R2: References: The author cites specific scientific works, enhancing the quality of the text. Including dates and sources would be helpful, especially considering the rapid progress in molecular biology.

Authors: We agree and thank the Reviewer 2 for the comments in the Reference list.

R2: Conclusion and Perspectives: The text concludes with a well-outlined summary, where the author emphasizes the importance of further research and the adaptation of legal regulations to advancing technology. This opens the door to future discussions, which is an asset.

Authors: We agree and thank the Reviewer 2 for the comments in the Conclusions.

R2: Overall Impression: The text is a solid exploration of the topic of genome editing in fruit trees. It introduces the reader to current achievements, challenges, and prospects in this fascinating field of plant biotechnology.

Authors: We agree and thank the Reviewer 2 for their editorial comments.

 We deeply again appreciate the efforts of the reviewer in the improvement of the manuscript for a future publication. This acknowledgement has been incorporated to the Acknowledgments section in page 17.

Yours faithfully,

Dr. Pedro Martínez-Gómez

CEBAS-CSIC, Murcia (Spain)

Reviewer 3 Report

Comments and Suggestions for Authors

This review included a synthesis of the relevant and current literature for the application of methods used to introduce CRISPR/Cas systems into fruit species, as well as the benefits and challenges of this gene editing technology. In addition, this review contains the latest news on legislation and regulations concerning the use of plants modified by CRISPR/Cas systems.

The introduction is well considered, accurately reflecting the current state of knowledge. The information included in this analysis is presented in a clear and well-structured manner.

The study is fully justified and motivated, as it is extremely important to improve transgenic research and make further efforts to increase the efficiency of regeneration and transformation procedures.

The development of CRISPR/Cas technology has opened a new perspective on the regulation of Genetically Modified Organisms (GMOs). This work fills some gaps in the impact of CRISPR/Cas on modifying plant architecture, improving fruit quality and yield, and tolerance to biotic and abiotic stresses.

The work is well organized, the Introduction is followed by the Mechanism of CRISPR/Cas system, Genetic transformation technology in fruit trees, CRISPR/Cas-mediated gene knock-in and knock-out in fruit trees, Off-target issue, Regulatory limits of genome editing, and Conclusions.

All parts are covered in depth, comprehensively described and substantial information is provided at a high level.

The conclusions contain some important key findings derived from the information provided as well as directions that need to be addressed in the future to establish efficient regeneration and transformation protocols for fruit trees species.

Bibliographical references are relevant to the topic and adequate for the accurate presentation of previous studies.

The paper is scientifically sound, based on clearly presented reasoning and not misleading.

Author Response

Dear Ms. Kittiya Kanngan, Assistant Editor of International Journal of Molecular Sciences,

please find enclosed the manuscript ijms-2710889 R1 entitled “CRISPR as genome editing technique in fruit trees breeding” which is the revised version of the former manuscript ijms-2710889 with the same title, which we would like to publish in your journal.

According with the suggestions of the Reviewer 3 we have revised the manuscript incorporating the proposed revisions indicating these revisions IN YELLOW.

We deeply appreciate the efforts of the reviewer in the improvement of the manuscript for a future publication.

Regarding Reviewer's 3 comments (R3):

R3: This review included a synthesis of the relevant and current literature for the application of methods used to introduce CRISPR/Cas systems into fruit species, as well as the benefits and challenges of this gene editing technology. In addition, this review contains the latest news on legislation and regulations concerning the use of plants modified by CRISPR/Cas systems.

Authors: We agree and thank the Reviewer 3 for their comments about the interest of this work and for considering this manuscript suitable for future publication.

R3: The introduction is well considered, accurately reflecting the current state of knowledge. The information included in this analysis is presented in a clear and well-structured manner.

Authors: We agree and thank the Reviewer 3 for their comments about the elaboration of the Introduction section.

 R3: The study is fully justified and motivated, as it is extremely important to improve transgenic research and make further efforts to increase the efficiency of regeneration and transformation procedures.

Authors: We agree and thank the Reviewer 3 for their comments about the interest of this work.

R3: The development of CRISPR/Cas technology has opened a new perspective on the regulation of Genetically Modified Organisms (GMOs). This work fills some gaps in the impact of CRISPR/Cas on modifying plant architecture, improving fruit quality and yield, and tolerance to biotic and abiotic stresses.

Authors: We agree and thank the Reviewer 3 for their editorial comments.

R3: The work is well organized, the Introduction is followed by the Mechanism of CRISPR/Cas system, Genetic transformation technology in fruit trees, CRISPR/Cas-mediated gene knock-in and knock-out in fruit trees, Off-target issue, Regulatory limits of genome editing, and Conclusions.

Authors: We thank the Reviewer 3 for their comments about the elaboration of the manuscript.

R3: All parts are covered in depth, comprehensively described and substantial information is provided at a high level.

Authors: We agree and thank the Reviewer 3 for their comments about the elaboration of the manuscript.

R3: The conclusions contain some important key findings derived from the information provided as well as directions that need to be addressed in the future to establish efficient regeneration and transformation protocols for fruit trees species.

Authors: We agree and thank the Reviewer 3 for their comments about the elaboration of the Conclusion section.

R3: Bibliographical references are relevant to the topic and adequate for the accurate presentation of previous studies.

Authors: We agree and thank the Reviewer 3 for their comments about the used References.

R3: The paper is scientifically sound, based on clearly presented reasoning and not misleading.

Authors: We agree and thank again the Reviewer 3 for their editorial comments.

 We deeply again appreciate the efforts of the reviewer in the improvement of the manuscript for a future publication. This acknowledgement has been incorporated to the Acknowledgments section in page 17.

Yours faithfully,

Dr. Pedro Martínez-Gómez

CEBAS-CSIC, Murcia (Spain)
